# Non-Coding RNA Related to MAPK Signaling Pathway in Liver Cancer

**DOI:** 10.3390/ijms231911908

**Published:** 2022-10-07

**Authors:** Qiuxia Wang, Jianguo Feng, Liling Tang

**Affiliations:** 1Key Laboratory of Biorheological Science and Technology, Ministry of Education, College of Bioengineering, Chongqing University, Chongqing 400044, China; 2Department of Anesthesiology, The Affiliated Hospital of Southwest Medical University, Luzhou 646000, China; 3Laboratory of Anesthesiology, Southwest Medical University, Luzhou 646000, China

**Keywords:** non-coding RNA, MAPK signaling pathway, liver cancer

## Abstract

The advancement in high-throughput sequencing analysis and the evaluation of chromatin state maps have revealed that eukaryotic cells produce many non-coding transcripts/RNAs. Further, a strong association was observed between some non-coding RNAs and cancer development. The mitogen-activated protein kinases (MAPK) belong to the serine–threonine kinase family and are the primary signaling pathways involved in cell proliferation from the cell surface to the nucleus. They play an important role in various human diseases. A few non-coding RNAs associated with the MAPK signaling pathway play a significant role in the development of several malignancies, including liver cancer. In this review, we summarize the molecular mechanisms and interactions of microRNA, lncRNA, and other non-coding RNAs in the development of liver cancer that are associated with the MAPK signaling pathway. Further, we briefly discuss the therapeutic strategies for liver cancer related to ncRNA and the MAPK signaling pathway.

## 1. Introduction

Hepatocellular carcinoma (HCC), intrahepatic cholangiocarcinoma (ICC), and combination hepatocellular carcinoma–cholangiocarcinoma (cHCC–CCA) are the three main pathophysiological subtypes of primary liver cancer. HCC accounts for 80–90%, and ICC accounts for 10–15% of all primary liver cancer cases [1]. In the past few years, there has been an increase in the incidences of primary liver cancer. In 2020, 830,000 deaths and 910,000 new incidences of primary liver cancer were reported worldwide [2]. HCC is the most prevalent type of primary liver cancer and the second leading cause of cancer-related mortality in males. It is also the sixth most common cancer and the third most common cause of cancer-related death worldwide [2]. Hepatitis B and C virus (HBV and HCV) infections, nonalcoholic hepatic adipose infiltration, alcohol consumption, and aflatoxin B1 exposure are the most common risk factors for HCC [3]. The treatment options available for liver cancer include surgical and non-surgical treatments. The surgical treatment includes surgical resection [4] and liver transplantation [5]. The non-surgical treatment includes interventional therapy, chemoradiotherapy, immunotherapy, and molecular-targeted therapy. Despite the emergence of multifunctional and cutting-edge therapies, the overall survival of HCC patients is still unsatisfactory [6]. Several studies have suggested that the lack of an early diagnosis and the high tumor heterogeneity could be attributed to the poor overall survival of HCC patients [7]. Hence, exploring the biological characteristics of HCC and the identification of new molecular targets are necessary for improving the diagnosis and treatment of HCC patients.

Mitogen-activated protein kinases (MAPKs) belong to the serine–threonine kinase family. The MAPK signaling pathway is primarily involved in cell proliferation from the cell’s surface to the nucleus [8]. In eukaryotic cells, the MAPK signaling pathway is a common and well-conserved regulatory mechanism [9]. The extracellular signal-regulated kinases (ERK MAPK, Ras/Raf1/MEK/ERK), MAPK14 (p38 MAPK), and stress-activated protein kinases or c-Jun N-terminal (SAPK or JNK) subfamilies are the three main types of MAPKs [10]. The MAPK signaling pathways are stimulated in response to external triggers such as growth factors, mitogens, cytokines, and stress. MAPKs affect cell proliferation, differentiation, apoptosis, inflammation, and metabolism, thereby regulating the biological behavior of cells via various signaling pathways. The MAPK signaling pathways play a crucial role in many human diseases, including cancer. The MAPK signaling pathway is activated in many tumors, and its various components have been identified as oncogenes [11]. The role of MAPKs in cancer has gained the attention of researchers, and targeting the MAPK signaling pathway could be a candidate for cancer therapeutics [12]. Multiple studies have demonstrated a strong correlation between alterations in the MAPK signaling pathway and cancers, including colorectal cancer [13], pancreatic cancer [14], liver cancer [15], and melanoma [16]. Previous studies reported that mutations in the MAPK signaling pathway were observed in HCC patients [17].

The advent of high-throughput sequencing techniques and the elucidation of the chromatin state maps have helped identify several non-coding transcripts/RNAs produced by eukaryotic cells [18,19,20]. Based on the arbitrary length of 200 ribonucleotides, the non-coding RNAs (ncRNAs) are classified into long ncRNAs (lncRNAs) and small ncRNAs. NcRNA includes microRNA (miRNA), long non-coding RNA (lncRNA), circular RNA (circRNA), small nucleolar RNA (snoRNA), etc. The ncRNA mediates its effect by acting a scaffold for protein complexes to regulate signaling pathways, alter protein expression by regulating transcription or translation, and interact with specific proteins as signaling partners [21]. Mounting evidence has demonstrated an association between the dysregulation of ncRNA and multiple diseases. For example, miRNA and lncRNA are associated with multiple cancers and affect the characteristics of the disease [22,23,24,25]. Moreover, they are also related to various complex biological processes, including the activity and functions of immune cells, immunological disorders [26], neurodevelopment, and neurological diseases [27]. Multiple studies have shown that a few ncRNAs are closely associated with the occurrence and progression of cancer and are clinical therapeutic targets. Some functionally important lncRNAs are lncRNA *HOX* transcript antisense RNA (HOTAIR), lncRNA highly upregulated in liver cancer (HULC), and telomerase RNA component (TERC). HOTAIR regulates the epigenetic silencing of *HOXD* sites [28], HULC acts as a competing endogenous RNA (ceRNA), and TERC is part of the catalytic center of the telomerase complex [29]. In addition, strategies to restore miRNA function have resulted in the development of MRX34 as an miRNA replacement therapy, which is currently undergoing clinical trials. MRX34 is a 120 nm liposome-comprising *MIR-34* simulant used to treat patients with metastatic or advanced liver cancer by intravenous injection [30]. Together, these results indicate a strong correlation between ncRNA and multiple cancers, including liver cancer. 

Mounting evidence has demonstrated a close correlation between the MAPK signaling pathway, ncRNA, and liver cancer. Further, ncRNA plays a significant role in the occurrence and progression of liver cancer. In this review, we summarize the molecular mechanisms of miRNA, lncRNA, and other ncRNAs associated with the MAPK signaling pathways in the progression of liver cancer, as well as their similarities and roles. We hope this review can shed light on the complex molecular network of the occurrence and development of HCC, and it could serve as the basis for developing therapeutic targets for liver cancer patients.

## 2. MicroRNA Related to the MAPK Signaling Pathway in Liver Cancer

To silence gene expression, the miRNAs, small ncRNAs with 22 nucleotides, drive the Argonaute (AGO) protein to the target of the 3’ untranslated regions (UTR) of mRNA [31]. MAPK signaling is typically mediated by the ERK, JNK, and p38 MAPK pathways. Studies have demonstrated the significance of the Ras–Raf–ERK signaling pathway in the growth-factor-mediated cell proliferation, differentiation, malignant transformation, and apoptosis of diverse cell types. The JNK family is primarily involved in the cellular stress response to radiation, osmolarity, and temperature changes. In addition, the p38 MAPK signaling pathway mediates inflammation, apoptosis, and other responses.

### 2.1. MicroRNA Related to the ERK Signaling Pathway

One of the most prominent kinase cascades in tumor cell biology is the Ras–Raf–MEK–ERK signaling pathway (also known as the Ras–ERK pathway) [32]. In mammalian cells, intracellular signaling pathways associated with ERK are considered the classical MAPK signaling pathway. Studies have confirmed that receptor tyrosine kinases (RTKs), G-protein-coupled receptors (GPCRs), and certain cytokine receptors activate the ERK signaling pathway. The ERK signaling pathway comprises two signaling axes that mediate the effect of MAPK on biological behaviors such as cell growth, differentiation, and apoptosis: the Src/Ras/MEKK2,3/MEK5/ERK5 and Ras/Raf/MEK1,2/ERK1,2 signaling pathways. Recent studies have suggested that multiple miRNAs are involved in the ERK signaling pathway, which affects the proliferation, migration, and invasion of liver cancer cells, thus regulating the occurrence, progression, and drug resistance in liver cancer.

Some miRNAs alter the expression of the ERK signaling pathway and vice versa, thus affecting the occurrence and progression of liver cancer. Certain microRNAs regulate the expression of Ras, which is involved in the MAPK signaling pathway.

A study by Yang et al., showed a significant decrease in miR-140-5p levels in all six liver cancer cell lines and HCC tissues [33]. Further studies have shown that miR-140-5p targets transforming growth factor-β receptor 1 (TGFBR1) and fibroblast growth factor 9 (FGF9). Abdel-Rahman et al. have demonstrated that FGF9 activates ERK/MAPK intracellular effectors [34]. Therefore, miR-140-5p regulates levels of endogenous ERK/MAPK-pathway-associated proteins such as H-RAS and p-ERK, which prevents the HCC cell proliferation and metastasis. In addition, Shi et al. showed a reduction in miR-296-5p levels in HCC tissues. The inhibition of the NRG1/ERBB2,3/RAS/MAPK/fra2 signaling pathway by targeting NRG1 attenuates epithelial–mesenchymal transition (EMT), thereby preventing HCC metastasis [35].

Apart from Ras, some microRNAs regulate the progression of liver cancers by targeting other proteins that regulate ERK1/2 phosphorylation.

Qin et al. [36] have shown that HBV suppresses the nuclear factor of activated T-cells 5 (NFAT5) expression by inducing the hypermethylation of activator protein 1 (AP1)-binding sites on the NFAT5 promoter in hepatoma cells. HBV also inhibits NFAT5 expression via miR-30e-5p by targeting *MAP4K4*, which phosphorylates ERK1/2 by acting as a mediator for MEKK1 phosphorylation, thereby inhibiting NFAT5 expression. The decrease in NFAT5 expression inhibits NFAT5 binding to the aspartyl-tRNA synthetase 2 (*DARS2*) promoter, thereby promoting *DARS2* (HCC oncogene) expression. This enhances cell cycle progression and inhibits the apoptosis of HCC cells. In vitro and in vivo studies have revealed that miR-320a acts as a tumor suppressor. MiR-320a interacts with PBX homeobox 3(*PBX3*) and increases the expression of matrix metallopeptidase 2 (MMP2) and cyclin-dependent kinase 2 (CDK2) by hindering the phosphorylation of ERK1/2, thus increasing the proliferation and metastasis of HCC cells [37]. Qin et al. [38] used RNA sequencing to demonstrate the expression profiles of circRNAs. The results revealed an association between circASAP1 and pulmonary metastasis in HCC patients following curative resection. Further, circASAP1 is derived from exons 2 and 3 of *ASAP1* (hsa_circ_0085616). CircASAP1 controls the miR-326/miR-532-5p-MAPK1 pathway by sponging miR-326 and miR-532-5p expression by the ceRNA mechanism. This promotes the proliferation and invasion of HCC cells. Recent studies have suggested low miR-198 expression in HCC compared to healthy liver parenchyma. A study by Tan et al. [39] revealed that miR-198 directly targets 3’UTR of cellular-mesenchymal epithelial transition factor (c-MET). High miR-198 expression decreases c-MET expression, which inhibits hepatocyte growth factor (HGF)-induced activation of ERK1/2 MAPK signaling in HCC cells, thereby inhibiting the migration and invasion of HCC cells.

### 2.2. MicroRNA Related to the JNK Signaling Pathway

The c-Jun N-terminal kinase (JNK), also known as stress-activated protein kinase, is a part of the MAPK family [40]. A few studies have reported the involvement of some microRNAs in the JNK pathway, which plays a significant role in liver damage, occurrence, and the development of liver cancer.

In the study by Deng et al. [41], the MAPK signaling pathway was triggered, c-Jun’s expression was increased, and the MAPK signaling pathway was activated in HCC cells infected with HCV. MiR-148a-3p targets c-Jun mRNA, thereby inhibiting the synthesis of the c-Jun protein, which prevents the activation of the MAPK signaling pathway. This suppresses the proliferation of HCV-infected HCC cells. Wang et al. [42] suggested that miR-1204 acts as an oncogene. MiR-1204 promotes cell proliferation and inhibits apoptosis via the MAPK and c-Jun signaling pathways by targeting zinc-finger protein 418 (ZNF418). Additionally, miR-125a-5p acts as a tumor suppressor in various malignancies, including HCC [43]. The miR-125a-5p attenuates the activation of the JNK signaling pathway by targeting protein tyrosine phosphatase N1 (PTPN1) and MAP3K11, thus preventing the growth of HCC cells and inducing apoptosis. 

### 2.3. MicroRNA Related to the p38 MAPK Signaling Pathway

P38 MAPK is the most important member of the MAPK family for regulating the inflammatory response. The p38 MAPK signaling pathway is triggered by physiological stress, lipopolysaccharide, osmotic stress, and UV irradiation. In liver cancer, miRNAs involved in the p38 MAPK signaling pathway regulate some biological behaviors of liver cancer cells.

Some miRNAs regulate the behavior of HCC cells by a direct association with the p38 MAPK signaling pathway. Chu et al. [44] revealed that an increase in the levels of lncRNA HULC attenuates p38 MAPK signaling and nuclear factor kappa B (NF-κB) in ATDC5 cells. This protects ATDC5 cells against tumor necrosis factor-α (TNF-α)-induced inflammatory damage by downregulating miR-101. Song et al. showed that polycyclic aromatic hydrocarbons (PAHs) are carcinogenic to humans [45]. Some studies showed that the miR-181 family targets MKP-5 to enhance the activation of the p38 MAPK signaling pathway, which plays a crucial role in PAH-induced hepatocarcinogenesis [46]. In addition, Xu et al. [47] showed a significant decrease in the levels of circSETD3 (hsa_circRNA_0000567/hsa_circRNA_101436) in HCC cell lines and tissues. Furthermore, circSETD3 sponges miR-421, thus confirming that MAPK14 (p38) is a target of miR-421. Therefore, it is tempting to postulate that circSETD3 inhibits the development of HCC via the circSETD3/miR-421/p38 pathway.

Some miRNAs affect HCC development by an indirect association with the p38 MAPK signaling pathways. According to Liu et al. [48], in HCC cells, the epidermal growth factor receptor (EGFR)–p38 MAPK axis increases the expression of programmed death-ligand 1 (PD-L1) via miR-675-5p and decreases the expression of human leukocyte antigen class-I (HLA-ABC) via hexokinase-2 (HK2). This study identified a new mechanism associated with immune suppression in HCC signaling networks. Yang et al. [49] revealed that miR-17-5p stimulates the p38 MAPK signaling pathway in human HCC cell lines, which increases the phosphorylation of heat shock protein 27, thereby promoting the migratory ability of HCC cells. Additionally, Wang et al. [50] demonstrated an association between a significant decrease in hsa_circ_0110102 expression in HCC cells and tissues and poor patient prognosis. CircRNA hsa_circ_0110102 sponges miR-508-5p, which prevents miR-508-5p from directly binding to the 3’UTR of peroxisome proliferators-activated receptor (PPAR), thereby promoting the synthesis and release of C-C chemokine ligand 2 (CCL2). Furthermore, CCL2 interacts with the cyclooxygenase-2/prostaglandin E2 (COX-2/PGE2) pathway in a p38-MAPK-dependent manner via forkhead box O1 (*FOXO1*), which inhibits the release of proinflammatory cytokines and HCC progression. Together, these results indicate that several miRNAs regulate the expression of several molecules via the p38 MAPK signaling pathway and indirectly affect HCC progression.

Based on these studies, it can be found that some microRNAs can participate in a separate MAPK pathway to affect HCC progression (Figure 1).

### 2.4. MicroRNA Related to Multiple MAPK Pathways

In addition to the previously mentioned miRNAs, which target a specific MAPK signaling pathway, some miRNAs simultaneously target multiple MAPK signaling pathways and participate in the complex regulatory network of HCC development. Some of the miRNAs and their associated mechanisms are described as follows:

Wang et al. [51] showed that miR-302a inhibits cell proliferation and promotes the apoptosis of human HCC cells by targeting MAP3K2 and PBX3 to reduce the expression of the key MAPK-associated proteins, including phosphorylated ERK (p-ERK1/2), p38 (p-p38), and JNK (p-JNK). Further, Tian et al. [52] demonstrated that a decrease in miR-520e expression was observed in HBV-positive HCC cells and tissues. Further studies have revealed that miR-520e overexpression specifically inhibited EPH receptor A2 (EphA2), thereby blocking the p38 MAPK and ERK1/2 signaling pathways. This results in HBV replication and the proliferation of HCC cells. Li et al. [53] showed a decrease in miR-129-5p levels in HCC cells and tissues. MiR-129-5p targets the calmodulin-dependent protein kinase IV (CAMK4) and decreases ERK, p38, and JNK protein phosphorylation, which inactivates the MAPK signaling pathway. This inhibits the proliferation, migration, and invasion of hepatocytes and promotes apoptosis. Interestingly, previous studies have shown that hepatic ischemia/reperfusion injury (IRI) is an inevitable pathological phenomenon in many pathological liver conditions [54,55]. Upon the activation of TNF-receptor-associated factor, 6miR-194 targets the pleckstrin homology-like domain family member 1 (PHLDA1) to inactivate p38, JNK, and apoptosis signal-regulating kinase 1 (ASK1). However, miR-194 does not alter the activation of ERK signaling, thereby improving stress and inflammation in hepatic IRI [56].

Additionally, some microRNAs associated with the MAPK signaling pathway regulate the occurrence and progression of liver cancer. These microRNAs also alter drug resistance, thereby affecting liver cancer treatment via multiple mechanisms or compounds. 

The first are the microRNAs associated with drug resistance. Dietrich et al. [57] used bioinformatics analysis and reporter assays to show that microRNA-622 directly targets KRAS. The loss of microRNA-622 causes the dysregulation of KRAS, which affects the downstream RAF/ERK and the PI3K/AKT signaling pathways, leading to the malignant transformation of liver cells, sorafenib sensitivity, and resistance. In addition, Azumi et al. [58] demonstrated that miR-181a targets an MAPK signaling factor, RASSF1. Further, the knockdown of RASSF1 expression increases resistance to sorafenib. 

Further, various compounds stimulate miRNA expression. Autotaxin, a secretory ectonucleotide enzyme, metabolizes lysophosphatidic choline to produce lysophosphatidic acid, which is distributed throughout the body. In addition to phospholipase activity, autotaxin, a phosphodiesterase, also influences inflammation and cancer progression [59,60,61]. Kuppa et al. [62] showed that miR-489-3p targets MEK1. Autotaxin suppresses miR-489-3p expression and enhances MEK1 expression, thus exacerbating tumor progression. Additionally, compounds such as Astragalus and Salvia miltiorrhiza extracts, often known as CASE, are considered active ingredients in Astragalus membranaceus. Wu et al. [63] showed that CASE switches pSmad3C signaling by upregulating miR-145 expression and promoting miR-145-regulated pSmad3L signaling. This reduces miR-21 expression and attenuates miR-21-regulated MAPK-dependent Smad3L signaling. CASE inhibits HCC progression via miR-21/Smad3L pathways in HepG2 cells and xenograft tumors from nude mice treated with TGF-β1. A naturally occurring flavonoid called luteolin (3,4,5,7-tetrahydroxy flavone) is found in various fruits and vegetables, including onion leaves, broccoli, carrots, parsley, chrysanthemum flowers, and sweet bell peppers [64]. Yang et al. [65] showed that luteolin upregulates miR-6809-5p levels. MiR-6809-5p targets flotillin 1 (FLOT1) and inhibits several downstream signaling pathways, including ERK1/2, p38, JNK, and NF-κB, to exert its biological effects, thus suppressing HCC cell proliferation. Hepatorenal toxicity from benzo[a]pyrene (BaP) is possible. Secoisolariciresinol diglucoside (SDG), a diphenolic molecule found in flaxseed, has anti-apoptotic, anti-inflammatory, and antioxidant properties and has been gaining more and more attention [66,67,68,69,70]. SDG targets the MIR101A/MKP-1 axis to downregulate p-ERK and p-p38 levels, thereby regulating oxidative stress and inflammation. This protected the kidney and liver of mice against BaP toxicity [71].

Based on these studies, it is evident that some microRNAs are simultaneously associated with multiple MAPK signaling pathways. This indicates that they do not affect a single pathway but have a holistic effect on the MAPK signaling network, thus regulating HCC development. It implies that miRNAs are closely related to the MAPK signaling network in HCC (Figure 2 and Table 1).

## 3. LncRNA Related to MAPK Signaling Pathway in Liver Cancer

### 3.1. LncRNA Related to the ERK Signaling Pathway

NcRNAs longer than 200 nucleotides are arbitrarily characterized as lncRNAs [72]. LncRNAs regulate various processes such as growth, differentiation, gene expression, and chromatin dynamics [18]. Recent studies have shown an association between multiple lncRNAs and the MAPK signaling pathway in liver cancer. They alter proliferation, migration, invasion, and drug resistance in liver cancer cells by interacting with DNA or proteins (Figure 3 and Table 2). We first discuss the lncRNAs involved in the ERK signaling pathway below:

Most lncRNAs related to the ERK signaling pathway promote liver cancer. Jiang et al. [73] used HepG2/ADR and MCF-7/ADR cells to study the differential expression and functions of a specific lncRNA termed Adriamycin Resistance Associated (ARA). The silencing of ARA expression revealed that suppressing ARA decreases cell proliferation and eventually leads to cell death, G2/M arrest, and abnormalities in cell migratory potential. The microarray results showed that ARA regulates various signaling pathways, including MAPK (ERK) signaling, metabolism, cell adhesion, and cell-cycle-related biochemical pathways. Further, ARA also regulates several cellular activities, including transcription and protein binding. A study by Xu et al. [74] identified a novel lncRNA called URHC. The expression of URHC was upregulated in HCC. URHC downregulates zipper-containing kinase expression by inhibiting the ERK/MAPK signaling pathway, thereby enhancing cell proliferation and inhibiting apoptosis. In addition, Wang et al. [75] revealed an increase in LINC01225 expression in HCC. LINC01225 binds to EGFR, which increases EGFR levels, and regulates the EGFR/Ras/Raf-1/MEK/MAPK signaling pathway, thus promoting the development and metastasis of HCC. Studies have revealed that of all the components of the MAPK signaling pathway, *MAPK6* is abundantly expressed by HCC samples [76]. Furthermore, a divergent lncRNA of MAPK6, known as lncMAPK6, is also overexpressed in liver cancer. LncMAPK6 interacts with the *MAPK6* promoter and recruits RNA polymerase II. This results in the activation of MAPK6 (ERK3) transcription, the enhancement of the MAPK signaling pathway, and the self-renewal of liver tumor-initiating cells via *MAPK6* expression. Previous studies have shown that an increase in the expression of lncRNA HOXD-AS1 was observed in HCC cell lines/tissues, which facilitates the metastasis of HCC cells by various mechanisms [86,87]. A study by Sun et al. [77] confirmed that lncRNA HOXD-AS1 has oncogenic properties. Silencing HOXD-AS1 expression attenuated the proliferation, migration, invasion, and cell cycle progression of HCC cells via the ERK signaling pathway. LL22NC03-N14H11.1 recruits c-MYB to suppress leucine zipper-like transcription regulator 1 (LZTR1) transcription, which hinders the LZTR1-regulated ubiquitination of H-RAS (G12V). This activates the MAPK signaling pathway involved in mitochondrial fission and p-DRP1 (Serine616) induction, thereby promoting HCC progression [78]. Moreover, H-RAS (G12V) activates the ERK signaling pathway, which participates in mitochondrial fission and p-DRP1 (S616) via ERK1/2, thereby promoting pancreatic cancer progression [88].

### 3.2. LncRNA Related to Other MAPK Signaling Pathways

In addition to the lncRNAs independently associated with the ERK signaling pathway, some lncRNAs associated with the p38 MAPK signaling pathway also affect the inflammatory response and metastasis in liver cancer.

As mentioned previously, in relation to miRNAs, increased levels of lncRNA HULC in HCC attenuates NF-κB and p38 MAPK signaling pathways in TNF-treated ATDC5 cells by downregulating microRNA-101. This protects ATDC5 cells against TNF-α-induced inflammatory damage [44]. Furthermore, lncRNA H19 binds to protein phosphatase 1 catalytic subunit alpha (PPP1CA) to dephosphorylate p38 MAPK, and subsequently downregulates osteoprotegerin expression, thus promoting HCC bone metastasis [79].

Some lncRNAs are not solely associated with a specific MAPK signaling pathway in primary liver cancers but can be related to several MAPK signaling pathways for regulating the progression of liver cancer. Some lncRNAs regulate cell proliferation to influence HCC progression. A study by Xu et al. [80] reported that two frequently downregulated lncRNAs in HCC had tumor inhibitory roles. Their expression was epigenetically inhibited by the histone methyltransferase enhancer of zeste homologue 2 (EZH2). First, EZH2 inhibits lncRNA TCAM1P-004 and decreases the binding of the lncRNA to insulin-like growth factor 2 mRNA-binding protein 1 (IGF2BP1) and H1.2 linker histone (HIST1H1C). Next, EZH2 inhibits lncRNA RP11-598D14.1, which decreases the binding of the lncRNA to IGF2BP1 and staufen double-stranded RNA binding protein 1 (STAU1). Together, these responses decrease the expression of the MAPK-pathway-associated genes such as *DDIT3*, *DUSP4*, and *RAP1A* and regulate the MAPK, p53, and HIF1α signaling pathways to promote HCC cell proliferation. A previous study has shown that RAP1A stimulates the ERK, p38 MAPK, and Notch signaling pathways, which induces the metastasis of ovarian cancer cells [89]. Therefore, it is hypothesized that the ERK/p38 signaling pathway is implicated in this. Furthermore, Pang et al. [81] demonstrated the binding of various lncRNAs to RPS6 in cancer cells, including LINC00998, which encodes for a short endogenous peptide known as SMIM30. It regulates cell proliferation and migration to promote the development of HCC by binding to the non-receptor tyrosine kinase SRC/YES1. This drives membrane anchoring and the phosphorylation of SRC/YES1 to activate the downstream ERK1/2 and p38 MAPK signaling pathways.

Some lncRNAs regulate apoptosis. A previous study has shown an increase in lncRNA HCG11 levels in HCC, which inhibits apoptosis and promotes HCC progression. LncRNA HCG11 mediates this effect by interacting with IGF2BP1, which interferes with the phosphorylation of the anti-apoptotic factors ERK, JNK, and p38 [82]. The inhibition of lncRNA HULC improves liver fibrosis and decreases the apoptosis of hepatocytes in rats with nonalcoholic fatty liver disease by inhibiting the p38 MAPK and JNK signaling pathways [83]. In HCC, an increase in the levels of linc00601 promotes HCC development by activating the ERK and p38 MAPK signaling pathways [84]. Further, a significant decrease in lncRNA RUNX1-IT1 expression was observed in HCC tissues [85]. Studies have shown that lncRNA RUNX1-IT1 could suppress the apoptosis of HCC cells and increases the expression of apoptosis-related proteins such as p-p38, p-ERK, and p-JNK.

## 4. Other Non-Coding RNA Related to the MAPK Signaling Pathway in Liver Cancer

In addition to microRNAs and lncRNAs, circRNAs associated with the MAPK signaling pathway also influence primary liver cancer. CircRNAs are produced by the back-splicing of protein-coding mRNAs or linear ncRNAs, connecting the upstream 3′ and downstream 5′ splice sites, forming a covalently closed continuous loop [90].

Some circRNAs associated with the MAPK pathway inhibit HCC. CircSETD3 (hsa_circRNA_0000567/hsa_circRNA_101436) is a novel HCC tumor suppressor. CircSETD3 sponges miR-421 via the ceRNA mechanism, which prevents the miR-421-induced degradation of MAPK14 (p38 MAPK), thus inhibiting HCC progress via the circSETD3/miR-421/MAPK14 pathway [47]. Wang et al. [50] showed that circRNA hsa_circ_0110102 sponges miR-580-5p to inhibit the binding of miR-580-5p to the 3’ UTR of PPAR and promotes the production and release of CCL2. Along with FoxO1, CCL2 promotes the COX-2/PGE2 pathway in a p38-MAPK-dependent manner, thereby inhibiting the release of pro-inflammatory cytokines by macrophages and HCC progression. Some circRNAs associated with the MAPK pathway promote HCC progression. Hu et al. [38] performed circRNA sequencing, and the results revealed an association between circASAP1 and lung metastasis in HCC patients following radical resection. CircASAP1 functions as a rival endogenous RNA to miR-532-5p and miR-326. In association with the miR-532-5p/miR-326-ERK2 signaling pathway, circASAP1promotes the proliferation and invasion of HCC cells. Additionally, Yuan et al. [91] showed an association between various differentially expressed circRNAs in HBV-positive liver cancer cells and the MAPK and Hippo signaling pathways. Of note, hsa_circ_0066966 is upregulated in HBV-positive liver cancer cells. Furthermore, hsa_circ_0066966 suppression significantly attenuates the proliferation and migration of liver cancer cells. A significant increase in hsa_circ_0002124 expression was observed in HCC cells and tissues [92]. The overexpression of hsa_circ_0002124 promotes the synthesis of p-ERK/ERK, p-p38, and p-JNK/JNK, thereby enhancing cell proliferation, invasion, and migration. It also inhibits apoptosis and cell cycle progression. 

In addition to circRNA, small nucleolar RNA (snoRNA) also associated with the MAPK signaling pathway influences HCC. SnoRNA is a class of extensively studied small ncRNAs and participates in the chemical modification of transfer RNAs (tRNAs), ribosomal RNAs (rRNAs), and small nuclear RNAs (snRNAs) to improve RNA folding [93]. Xu et al. [94] showed significant dysregulation in the expression of SNORD113-1 in HCC tumors compared to the surrounding non-tumor tissues. Further studies have revealed that SNORD113-1 attenuates the phosphorylation of SMAD2/3 and ERK1/2 in the ERK/MAPK and TGF-β pathways, thus inhibiting tumorigenesis in HCC.

One of the most extensively studied lncRNAs is the metastasis-associated lung adenocarcinoma transcript 1(MALAT1) [95]. The nuclear-localized MALAT1 is the source of the MALAT1-associated small cytoplasmic RNA (mascRNA), a cytoplasmic tRNA-like short RNA [96,97,98]. Xie et al. [99] discovered an increase in levels of mascRNA and its parental lncRNA MALAT1 in HCC samples and various cancer cell lines. This increase in the levels of mascRNA and lncRNA MALAT1 promotes the proliferation and metastasis of HCC cells by activating the ERK signaling pathway.

Further, vault RNA (vtRNA) is a novel family of short non-coding RNAs [100]. VtRNA is a component of a hollow barrel ribonucleoprotein (RNP) complex. They are large 13-MDa vault particles of 80–150 nucleotides in length. Ferro et al. [101] demonstrated that the knockout of VTRNA1-1 in HCC cells increases the activation of the MAPK signaling pathway. ERK2-ERK1 is responsible for transcription factor EB (TFEB) inactivation and the cytoplasmic retention of TFEB. The silencing of VTRNA1-1 suppresses the nuclear localization of TFEB and decreases the expression of genes associated with the coordinated lysosomal expression and regulation (CLEAR) network. Further, it significantly reduces the lysosomal compartment dysfunction in in vivo and in vitro tumor cell proliferation.

Together, these results suggest that other non-coding RNAs contribute to the development of liver cancer via the MAPK signaling pathway (Table 3).

## 5. Discussion

Despite the advancement in therapeutics, the overall survival rates of HCC patients are still low. This could mainly be due to a lack of early diagnosis and the high heterogeneity of tumors. Hence, it is crucial to develop new molecular targets. A close association between the MAPK signaling pathway and liver cancer has been established, and ncRNA also plays a significant role in the progression of liver cancer. Hence, the association between ncRNA, the MAPK signaling pathway, and liver cancer should be extensively analyzed. In this review, we summarized the molecular mechanisms of microRNA, lncRNA, and other ncRNAs in the occurrence and development of primary liver cancer associated with the MAPK signaling pathway. Our analysis showed that some ncRNAs are associated with a specific MAPK signaling pathway, which participates in the development of liver cancer by regulating the expression or phosphorylation of specific proteins. Further, some ncRNAs are extensively involved in several MAPK signaling pathways, regulating the expression of proteins associated with the ERK, p38 MAPK, and JNK signaling pathways, which affect the progression of liver cancer. Furthermore, some ncRNAs are also stimulated by some compounds that affect the MAPK signaling pathway or are involved in drug resistance by regulating the MAPK signaling pathway.

With the rapid development of sequencing and bioinformatics technologies, various ncRNAs are constantly being discovered, some of which play a significant role in cancer development. Hence, drugs targeting these ncRNAs can be used for treating cancer. For example, the first miRNA drug, miR-122 antagonist Miravirsen, has entered phase II clinical trials [102]. Therefore, it is worth exploring whether these ncRNAs that are related to the MAPK signaling pathway can be used as therapeutic targets or biomarkers in liver cancer.

Additionally, significant developments have been made in exploring ncRNA in recent years. A few studies have demonstrated that some ncRNAs can be translated. As mentioned earlier, the peptide SMIM30 encoded by lncRNA LINC00998 regulates cell proliferation and migration to promote the progression of HCC. SMIM30 influences HCC progression by binding to SRC/YES1, which drives its phosphorylation and membrane anchoring, promoting the downstream ERK1/2 and p38 MAPK signaling pathways [81]. In addition, lncRNA was also found to encode a small stretch of peptide, which is vital in regulating cancer progression. For example, the lncRNA-encoded polypeptide ASRPS suppresses angiogenesis in triple-negative breast cancer (TNBC) [103]. Small proteins in lncRNA LOC90024 promote tumorigenesis and RNA splicing in cancer [104]. Moreover, the micropeptide ASAP, encoded by LINC00467, accelerates the development of colorectal cancer by directly controlling ATP synthase activity [105]. These results indicate that additional studies should be conducted on exploring ncRNA, which could serve as therapeutic targets and diagnostic molecular markers. Furthermore, exosomal ncRNA plays a crucial role in the development of liver cancer. The macrophages in HCC target the 3’UTR of androgen receptor (AR) mRNA using exosome-encapsulated miR-92a-2-5p, which inhibits AR translation, modifies the PHLPP/p-AKT/-catenin signaling pathway and promotes HCC cell invasion [106]. The lncRNA HOTAIR modulates RAS-related protein RAB35 and synaptosome-associated protein 23 (SNAP23) to promote the secretion of exosomes by HCC cells [107]. HCC-derived exosomal circUHRF1 depletes natural killer cells, which could induce resistance to anti-PD-1 therapy [108]. Various studies have shown that exosomal ncRNAs circulating in the blood of HCC patients, such as miR-21 [109], miR-122 [110] and lncRNA-ATB [111], could independently be used as biomarkers for HCC staging, thereby evaluating therapeutic efficacy, and prognosis. Based on this evidence, it is clear that the development of liver cancer could be significantly influenced by a family of molecules known as exosomal ncRNAs. These exosomal ncRNAs could be used in the therapeutics, diagnosis, and management of HCC. 

NcRNAs target the MAPK signaling pathway and upstream and downstream target molecules in various cancers. Several upstream cytokines and membrane receptors activate the MAPK signaling pathway. In gastric cancer, miR-143 indirectly downregulates the expression of the human epidermal growth factor receptor (HER2), which is an upstream molecule of KRAS, by silencing DEAD/H-box RNA helicase 6 (DDX6) [112]. In breast cancers, miR-4500 exerts antitumor effects by inhibiting the MAPK signaling pathway via ribonucleotide reductase subunit M2 (RRM2) [113]. In the luminal B (LumB) subtype of breast cancer, lncRNA LIPR-AS1 cooperative genes *IL1R* and *TGFBR*, which are upstream of the MAPK signaling pathway, share a common grade junction pathway (p38 MAPKs-MEF2C), which induces the proliferation, differentiation and apoptosis of cells [114]. In addition, non-coding RNA also targets molecules downstream of the MAPK signaling pathway. In acute myeloid leukemia (AML), miR-193b arrests the cell cycle at the G1/S phase and induces apoptosis by targeting multiple factors and the downstream cell cycle regulator of the KIT-RAS-RAF-MEK-ERK (MAPK) signaling pathway [115]. MiR-155 regulates radiation-induced senescence by acting downstream of the p53 and p38 MAPK pathways [116]. In malignant pleural mesothelioma (MPM), miR-206 targets CDK6 and arrests the cell cycle at the G1/S phase [117].

The upstream RTKs receptors of MAPKs play a critical role in the activation of signaling pathways, including EGFR, hepatocyte growth factor receptor (HGFR; c-MET), Vascular Endothelial Growth Factor receptor (VEGFR), platelet derived growth factor receptor (PDGFR), etc. In HCC, these receptors are also regulated by some microRNAs. MiR-206 inhibits the expansion of liver cancer stem cells and HCC cell dedifferentiation by targeting the EGFR signaling pathway [118]. The miR-874, a negative regulator of δ opioid receptor (DOR), inhibits HCC proliferation and metastasis by targeting the DOR/EGFR/ERK pathway [119]. MiR-449 binds and reduces the levels of C-MET mRNA, which reduces proliferation, and promotes the apoptosis of hepatocytes [120]. MiR-206 targets c-Met and CDK6 to inhibit the development of HCC [121]. MiR-199a-3p suppresses VEGF secretion in cancer cells (CACs) and the expression of endothelial cells, VEGFR1 and VEGFR2, thereby limiting the interaction between CACs and endothelial cells [122]. LncRNA MYLK-AS1 directly targets miR-424-5p. The MYLK-AS1/miR-424-5p signaling pathway promotes angiogenesis and proliferation by VEGFR-2 signaling through E2F transcription factor 7 (E2F7) [123]. BRCA1 mediates miR-146a to increase PDGFRα (PDGFRA) expression in the human umbilical vein endothelial cells (HUVECs) [124]. LINC00467 sponges miR-509-3p, thereby upregulating PDGFRA expression in HCC cells and promoting the proliferation and invasion of HCC [125]. 

The MAPK signaling pathway activates the downstream target genes and induces transcription of various genes, including cell-cycle-associated genes such as CDK4/6. These genes are also regulated by microRNAs. In Osteosarcoma (OS) cells, miR-338-3p acts as a tumor suppressor by targeting runt-related transcription factor 2 (RUNX2) and CDK4, which inhibits the MAPK pathway [126]. In bladder cancer cells, miR-106a inhibits the expression of cyclin D1 and CDK6 and enhances the p21^CIP1/WAF1^ pathway, which arrests the cell cycle at the G1 phase [127]. These results indicate that ncRNA regulates molecules upstream and downstream of the MAPK signaling pathway. 

As mentioned earlier, miR-622 [57] and miR-181a [58], which are associated with the MAPK pathway, affect the resistance of sorafenib, used for liver cancer treatment. In cancer therapeutics, drug resistance is a major problem that involves complex mechanisms and is affected by multiple factors. The issue of drug resistance has attracted tremendous attention as tumor-targeted treatment continues to develop.

In 2010, Settleman et al. [128] detected a small subset of reversible "resistant" cells, called a subset of slowly circulating quiescent “drug-tolerant persisters” (DTPs) cells. Such cells require the histone demethylase RBP2/KDM5A/Jarid1A to maintain cell viability by participating in insulin like growth factor 1 receptor (IGF-1R) signaling and changing the state of chromatin. A fraction of DTPs begins to proliferate after continuous treatment with erlotinib, giving rise to the second cohort of cells labeled as “drug-tolerant expanded persisters” (DTEPs). In order to protect the cells from potentially lethal exposure to the drug, the cancer cell uses this dynamic survival mechanism, which presents a temporary state of drug resistance reversal. Various studies have performed single-cell RNA sequencing on tumors derived from patients, xenografts, and cell lines, including different subpopulations of cancer cells [129]. The results revealed that the combination of crizotinib and EGFR inhibitor inhibits the emergence of EGFR-inhibitor-tolerant clones in non-small-cell lung carcinoma cells. Additionally, the release of cytochrome c and mitochondrial outer membrane permeability (MOMP) are prerequisites for the development of a persistent phenotype in drug-resistant cells. Studies have shown that the cytochrome c-heme-regulated inhibitor (HRI)-activating transcription factor 4 (ATF4) pathway drives the persistent phenotype of the cells. ATF4 expression promotes downstream processes, such as metabolic reprogramming, cell cycle suppression, immune evasion, and the activation of EMT, which are essential for the persistent phenotype [130]. Drug resistance is often associated with mutations in genes targeted by the drugs; however, other mechanisms underlying drug resistance have also gradually been discovered. Drug resistance in prostate cancer is accompanied by cell lineage changes. Various studies have used human samples, transgenic mice, and organoid culture triple methods to study changes in cell lineage in prostate cancer. The results revealed that the lineage plasticity in prostate cancer originates from epithelial cells as defined by a mixed lumen–basal phenotype. Furthermore, the JAK/STAT signaling pathway and FGFR are the main drivers of transforming prostate cancer into castration-tolerant or castration-resistant prostate cancer (CRPC) [131].

A combination of inhibitors has been used to address the issue of drug resistance in cancer therapeutics. EGFR inhibitors are often utilized for the treatment of EGFR-mutant lung cancer patients; however, acquired drug resistance has reduced its efficacy. A study has shown that the combined inhibition of Aurora B kinase and EGFR prevents and eliminates the resistance to EGFR inhibitors in lung cancer by promoting BIM and PUMA-mediated apoptosis [132]. Several inhibitors have been developed to target different molecules of the MAPK signaling pathway. For example, in 2013, dabrafenib was approved as a B-Raf proto-oncogene (*BRAF*) inhibitor for treating BRAF-V600K-mutant metastatic melanoma [133]. As a MEK inhibitor, trametinib has been approved for the treatment of BRAF-V600E metastatic melanoma [134]. However, patients treated with RAF or MEK inhibitors frequently develop drug tolerance. On further investigation, it has been revealed that the use of ERK inhibitors could help overcome the acquired resistance due to the use of upstream kinase inhibitors. Studies have shown that selective ERK inhibitors can overcome resistance to both BRAF and MEK inhibitors, as well as acquired resistance to MEK inhibitors [135,136]. SCH772984 is one of the selective ERK inhibitors. Tumor cells resistant to BRAF inhibitors or MEK inhibitors can be sensitized by treatment with SCH772984. Therefore, the combination of ERK inhibitors and BRAF or MEK inhibitors could be used in cancer therapeutics [137,138,139]. Additionally, it has been demonstrated that the simultaneous inhibition of MEK and ERK can prevent the development of resistance and eliminate acquired resistance to MEK inhibitors. Ulixertinib (BVD-523) is a widely used ERK inhibitor for the treatment of NRAS-mutated melanoma and BRAF-mutated solid tumors [140]. The combined use of GDC-0994 and the MEK inhibitor cobimetinib demonstrated improved anticancer efficacy in KRAS- and BRAF-mutated tumor models [141]. This evidence indicates that a combination of MAPK inhibitors and other inhibitors could be an effective strategy to overcome drug resistance.

Together, these results suggest that the regulation of the MAPK signaling pathway and its upstream as well as downstream molecules by non-coding RNAs plays a vital role in the development, diagnosis and treatment of liver cancer. In this review, we only explored the non-coding RNAs associated with the MAPK signaling pathways in liver cancer. A molecular network is composed of multiple interrelated signaling pathways in the progression of HCC. Some non-coding RNAs simultaneously affect multiple signaling pathways, which play a more significant role in the occurrence and development of liver cancer. MAPK inhibitors are crucial for cancer therapeutics. Further, a combination of inhibitors targeting ncRNA and MAPK pathways could be a promising candidate in cancer therapeutics, especially in drug resistance (Figure 4).

## Figures and Tables

**Figure 1 ijms-23-11908-f001:**
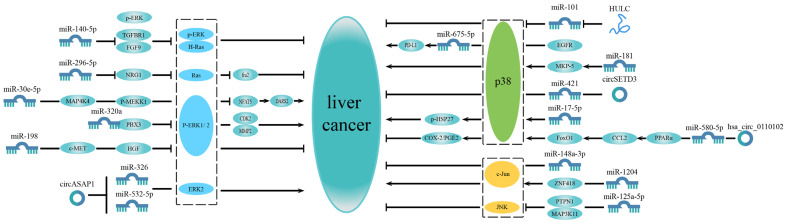
MicroRNAs related to a single MAPK pathway in liver cancer. The MAPK signaling pathway is divided into three parts: ERK signaling pathway, p38 MAPK signaling pathway and JNK signaling pathway. Some microRNAs only affect liver cancer progression by participating in the ERK signaling pathway, while some microRNAs are associated with the p38 MAPK signaling pathway to regulate liver cancer progression. Some microRNAs function via the JNK signaling pathway. Most of these microRNAs regulate the MAPK pathway upstream to affect liver cancer.

**Figure 2 ijms-23-11908-f002:**
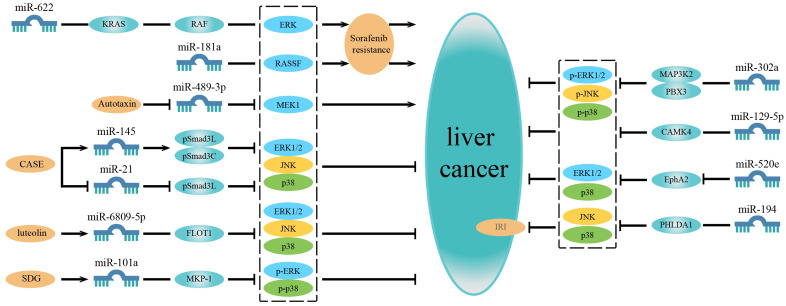
MicroRNAs related to multiple MAPK signaling pathways or regulated by compounds in liver cancer. On the left are microRNAs associated with drug resistance and microRNAs regulated by compounds. On the right are microRNAs simultaneously involved in multiple MAPK signaling pathways. In addition to the microRNAs affecting a single certain pathway, some microRNAs are associated with multiple MAPK signaling pathways and have a universal effect on the MAPK signaling network, thereby regulating the development of liver cancer. Besides, some microRNAs are related to sorafenib resistance or are regulated by some compounds to affect the progression of liver cancer.

**Figure 3 ijms-23-11908-f003:**
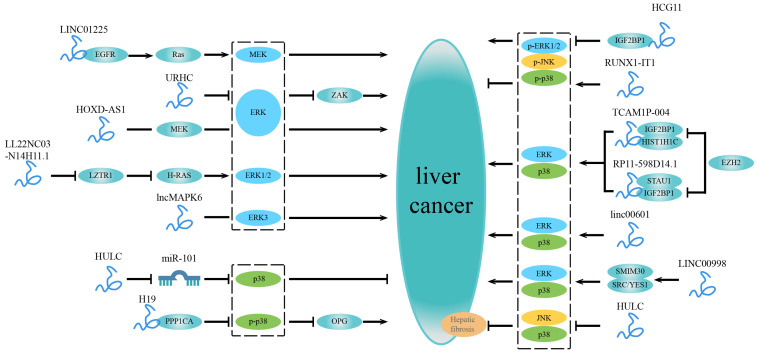
LncRNAs related to the MAPK signaling pathway in liver cancer. On the left are lncRNAs that are independently involved in a specific MAPK signaling pathway and impact the development of liver cancer. LncRNAs involved in multiple MAPK signaling pathways are shown on the right. Some lncRNAs can bind to specific factors or mediate a particular factor to regulate a certain MAPK signaling pathway. In contrast, other lncRNAs are widely involved in several MAPK signaling pathways, affecting the progression of liver cancer.

**Figure 4 ijms-23-11908-f004:**
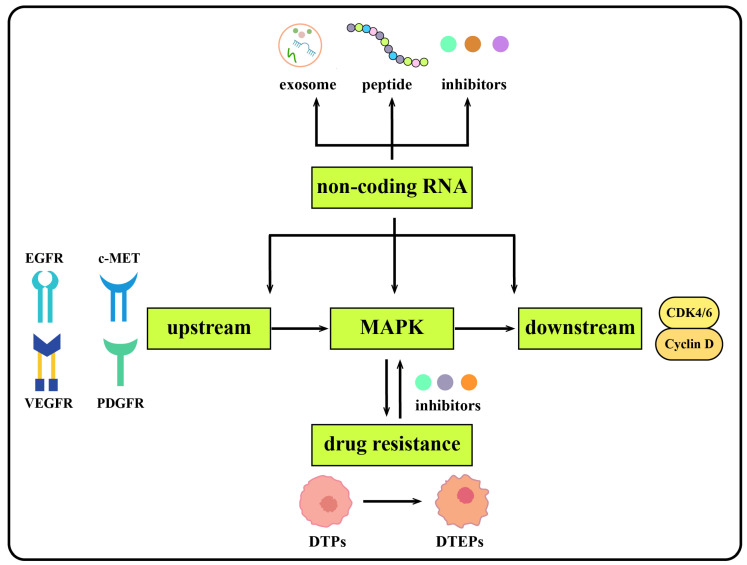
Graphic summary of the discussion.

**Table 1 ijms-23-11908-t001:** MicroRNAs related to the MAPK signaling pathway in liver cancer.

Gene Name	Expression Level in Liver Cancer	Promotion or Inhibition of Liver Cancer	MAPK Pathway	Key Factors in MAPK	Biological Functions	Reference
*miR-140-5p*	down	Inhibition	ERK	p-ERK, H-Ras	proliferation, metastasis	[34]
*miR-296-5p*	down	Promotion	ERK	RAS	metastasis	[35]
*miR-30e-5p*	up	Promotion	ERK	ERK1/2, MAP4K4, MEKK1	cell cycle, progression, apoptosis	[36]
*miR-320a*	down	Promotion	ERK	ERK1/2	EMT, proliferation, metastasis	[37]
*miR-326*	/	Inhibition	ERK	MAPK1	proliferation, invasion	[38]
*miR-532-5p*	/	Inhibition	ERK	MAPK1	proliferation, invasion	[38]
*miR-198*	down	Inhibition	ERK	ERK1/2	migration, invasion	[39]
*miR-148a-3p*	up	Inhibition	JNK	c-Jun	proliferation	[41]
*miR-1204*	up	Promotion	JNK	c-Jun	proliferation, apoptosis	[42]
*miR-125a-5p*	down	Inhibition	JNK	MAP3K11	proliferation, apoptosis	[43]
*miR-101*	down	Inhibition	p38MAPK	p-p38	inflammatory	[44]
*miR-181*	up	Promotion	p38MAPK	MKP-5	metastasis	[46]
*miR-421*	up	Promotion	p38MAPK	MAPK14	growth	[47]
*miR-675-5p*	down	Promotion	p38MAPK	/	Immune suppression	[48]
*miR-17-5p*	up	Promotion	p38MAPK	p-p38	migration	[49]
*miR-580-5p*	up	Promotion	p38MAPK	/	release of proinflammatory cytokines	[50]
*miR-302a*	down	Inhibition	ERK, p38MAPK, JNK	MAP3K2 p-ERK1 p-p38, p-JNK	proliferation, apoptosis	[51]
*miR-520e*	down	Promotion	ERK, p38MAPK	/	growth	[52]
*miR-129-5p*	down	Inhibition	ERK, p38MAPK, JNK	p-ERK1/2, p-p38, p-JNK	proliferation, migration, invasion, apoptosis	[53]
*miR-194*	down	Inhibition	p38MAPK, JNK	p-p38, p-JNK	inflammation	[56]
*miR-622*	down	Inhibition	ERK	KRAS	malignant development, sorafenib resistance	[57]
*miR-181a*	/	Promotion	ERK	RASSF	sorafenib resistance	[58]
*miR-489-3p*	up	Inhibition	ERK	MEK1	tumor progression	[62]
*miR-145*	down	Inhibition	ERK, p38MAPK, JNK	p-ERK1/2, p-p38, p-JNK	tumor progression	[63]
*miR-21*	up	Promotion	ERK, p38MAPK, JNK	p-ERK1/2, p-p38, p-JNK	tumor progression	[63]
*miR-6809-5p*	/	Inhibition	ERK, p38MAPK, JNK	p-ERK1/2, p-p38, p-JNK	growth	[65]
*miR-101a*	up	Inhibition	ERK, p38MAPK	MKP-1	oxidative stress inflammation	[71]

**Table 2 ijms-23-11908-t002:** LncRNAs related to multiple MAPK signaling pathways in liver cancer.

Gene Name	Expression Level in Liver Cancer	Promotion or Inhibition of Liver Cancer	MAPK Pathway	Key Factors in MAPK	Biological Functions	Reference
*ARA*	up	Promotion	ERK	MEKK 2	proliferation, Adriamycin resistance	[73]
*URHC*	up	Promotion	ERK	p-ERK	proliferation, apoptosis	[74]
*LINC01225*	up	Promotion	ERK	Ras, Raf-1, MEK	occurrence, metastasis	[75]
*lncMAPK6*	up	Promotion	ERK	MAPK6 (ERK3)	self-renewal of liver TICs	[76]
*HOXD-AS1*	up	Promotion	ERK	MEK	proliferation, cell cycle, migration, invasion	[77]
*LL22NC03-N14H11.1*	up	Promotion	ERK	H-RAS, ERK1/2	progression	[78]
*HULC*	up	Promotion	p38MAPK	p-p38	inflammatory	[44]
*H19*	up	Promotion	p38MAPK	p-p38	bone metastasis	[79]
*TCAM1P-004*	down	Inhibition	ERK, p38MAPK	p-p38, p-JNK	proliferation	[80]
*RP11-598D14.1*	down	Inhibition	ERK, p38MAPK	p-p38, p-JNK	proliferation	[80]
*LINC00998*	up	Promotion	ERK, p38MAPK	SRC	proliferation, migration	[81]
*HCG11*	up	Promotion	ERK, p38MAPK, JNK	p-ERK, p-p38, p-JNK	apoptosis, progression	[82]
*HULC*	up	Promotion	p38MAPK, JNK	p-ERK, p-p38, p-JNK	fibrosis, apoptosis	[83]
*linc00601*	up	Promotion	ERK, p38MAPK	p-ERK, p-p38,	progression	[84]
*RUNX1-IT1*	down	Inhibition	ERK, p38MAPK, JNK	p-ERK1/2, p-p38, p-JNK	apoptosis	[85]

**Table 3 ijms-23-11908-t003:** Other non-coding RNAs related to multiple MAPK signaling pathways in liver cancer.

Gene Type	Gene Name	Expression Level in Liver Cancer	Promotion or Inhibition of Liver Cancer	MAPK Pathway	Key Factors in MAPK	Biological Functions	Reference
circRNA	*CircSETD3*	down	Inhibition	p38 MAPK	MAPK14	growth	[47]
circRNA	*hsa_circ_0110102*	down	Inhibition	p38 MAPK	p38	pro-inflammatory cytokines release	[50]
circRNA	*circASAP1*	up	Promotion	ERK	ERK2	proliferation, invasion	[38]
circRNA	*hsa_circ_0066966*	up	Promotion	MAPK	/	proliferation, migration	[91]
circRNA	*hsa_circ_0002124*	up	Promotion	ERK, p38MAPK, JNK	p-ERK, p-p38, p-JNK	proliferation, migration, invasion, apoptosis	[92]
snoRNA	*SNORD113-1*	down	Inhibition	ERK	ERK1/2	tumorigenesis	[94]
tRNA-like small RNA	*mascRNA*	up	Promotion	ERK	ERK, p-ERK	proliferation, migration	[99]
vtRNA	*VTRNA1-1*	up	Promotion	ERK	ERK1/2	progression	[101]

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
