# Peer review of "Non-Coding RNA Related to MAPK Signaling Pathway in Liver Cancer"

_ijms, 2022, doi:10.3390/ijms231911908_

Round 1

Reviewer 1 Report

The submitted review article by Wang, et al., titledNon-coding RNA related to MAPK signaling pathway in liver canceris interesting. In this article, the authors have compiled the role of noncoding RNA in the regulation of the MAPK signaling pathway in liver cancer pathogenesis. I have some suggestions to improve the articles.

1.     If Authors are talking about regulation/targeting of MAPK (ERK, JNK and p38) signaling pathway by micro-RNA and long noncoding RNA, they should also consider targeting upstream regulators and downstream molecules (for example PMID: 33996251)

2.     MAPK upstream RTKs receptors like EGFR, MET, VEGFR, PDGFR etc targeting is very crucial for signaling. So, it would be good to include potential micro-RNAs which regulate these receptors in liver cancer.

3.     Also discuss the microRNAs that regulate MAPK downstream CDK4/CDK6 molecules. 

Author Response

Response to Reviewer 1

Dear Editor and Reviewers,

Thank you for your useful comments and suggestions on our manuscript. We have modified the manuscript accordingly as follow:

Comment 1: If Authors are talking about regulation/targeting of MAPK (ERK, JNK and p38) signaling pathway by micro-RNA and long noncoding RNA, they should also consider targeting upstream regulators and downstream molecules (for example PMID: 33996251)

Response: Thank you for this very insightful comment. In the revised Discussion section, we have added the regulation of microRNA and lncRNA on the upstream regulators and downstream target molecules of MAPK, and added your recommended reference (PMID: 33996251) and other relevant references. (line 490-506)

Comment 2: MAPK upstream RTKs receptors like EGFR, MET, VEGFR, PDGFR etc. targeting is very crucial for signaling. So, it would be good to include potential micro-RNAs which regulate these receptors in liver cancer.

Response: We gratefully appreciate for your valuable suggestion. In the modified Discussion section, we added some microRNAs that regulate RTKs receptors in liver cancer. (line 507-524)

Comment 3: Also discuss the microRNAs that regulate MAPK downstream CDK4/CDK6 molecules.

Response: Thank you for your sincere suggestions. In the revised Discussion, we have added the regulation of microRNAs on CDK4/6 associated with MAPK. (line 525-532)

Finally, we thank you once again for your valuable comments and suggestions. We deeply appreciate your consideration of our manuscript. If you have any queries, please do not hesitate to contact us.

Thank you and best regards.

Sincerely yours,

Liling Tang

PhD, Professor

Reviewer 2 Report

The text is worthwhile, but:

A. Minor remarks.

1) Grammar (especially plurals) must be corrected by a native English speaker, starting from the title; lapses like: "along with liver cancer...lncMAPK6 is also overexpressed" should be corrected (lines 299 are -not re; line 276, line 317, 324, 440, 497 and so on)

2) Figures 1-3: difficult to read even at 200% magnification (especially Fig.1), letters in rectangles must be converted into black ones (unreadable right now);

3) Table 1 heading/title: this is a Table, etc... Is it a joke, which should be printed?

B. Major points.

1) Discussion is weak and should be rewritten. E.g. in the text: miR140-5p down, hence TGFBR1 and FGF9 down. Good, but that means that cells undergo partial EMT and become DTEP cells, which are at the center of comprehensive picture of cancer cells drug resistance (20371346, 33712615, 36055199, 35981096).

2) I would add a bit about MAPK inhibitors, which overcome drug resistance (34388376, etc.) and would shorten the section on natural "cancer cures";

3) Actually, Tables don't add anything to the major line of logic; I recommend to convert them into Supplementary (it's just a glossary);

4) Finally, the crowning achievement of any good discussion is a concise figure that summarizes everything that's important. 

Author Response

Response to Reviewer 2

Dear Editor and Reviewers,

Thank you for your useful comments and suggestions on our manuscript. We have modified the manuscript accordingly as follow:

  1. Minor remarks.

Comment 1: Grammar (especially plurals) must be corrected by a native English speaker, starting from the title; lapses like: "along with liver cancer...lncMAPK6 is also overexpressed" should be corrected (lines 299 are -not re; line 276, line 317, 324, 440, 497 and so on)

Response: Thank you for your careful check. We apologize for the grammar and language issues presented in the review. We have carefully checked the entire manuscript for grammatical errors according to your suggestions and have sent the manuscript to Bullet Edits Company to improve the quality of the language. Please see the attachment.

Comment 2: Figures 1-3: difficult to read even at 200% magnification (especially Fig.1), letters in rectangles must be converted into black ones (unreadable right now);

Response: We gratefully appreciate for your meticulous check. We have changed the letters in the rectangle to black and re-uploaded high-quality clear images.

Comment 3: Table 1 heading/title: this is a Table, etc... Is it a joke, which should be printed?

Response: Thank you very much for your correction. We apologize for the incorrect title of Table 1 in the manuscript due to insufficient checking. We have corrected it as: Table1. MicroRNAs related to MAPK pathway in liver cancer. (line 292)

  1. Major points.

Comment 1: Discussion is weak and should be rewritten. E.g. in the text: miR140-5p down, hence TGFBR1 and FGF9 down. Good, but that means that cells undergo partial EMT and become DTEP cells, which are at the center of comprehensive picture of cancer cells drug resistance (20371346, 33712615, 36055199, 35981096).

Response: We gratefully appreciate for your valuable suggestion. We have rephrased miR-140-5p in the revised manuscript. In addition, we have read the paper you recommended and added the discussion on drug resistance of cancer cells in the modified manuscript. (line 124-126, line 533-566)

Comment 2: I would add a bit about MAPK inhibitors, which overcome drug resistance (34388376, etc.) and would shorten the section on natural "cancer cures";

Response: Thank you very much for your suggestion and recommended papers. We have supplemented the discussion on MAPK inhibitors in the revised manuscript. In addition, for the sake of completeness of our view, we have kept the section on natural "cancer cures". (line 567-591)

Comment 3: Actually, Tables don't add anything to the major line of logic; I recommend to convert them into Supplementary (it's just a glossary);

Response: Thank you very much for your sincere suggestion. The purpose of our tables is to summarize the non-coding RNAs discussed in this manuscript, so that readers can see the overview of each non-coding RNA more concisely and clearly. Furthermore, considering this is a review, we do not know if the journal would be willing to put these tables in the supplementary.

Comment 4: Finally, the crowning achievement of any good discussion is a concise figure that summarizes everything that's important.

Response: Thank you for your valuable and constructive comment. We have organized the revised discussion section into a concise schematic according to your suggestion.

Finally, we thank you once again for your valuable comments and suggestions. We deeply appreciate your consideration of our manuscript. If you have any queries, please do not hesitate to contact us.

Thank you and best regards.

Sincerely yours,

Liling Tang

PhD, Professor

Round 2

Reviewer 1 Report

The authors have successfully addressed my concerns. The article is acceptable.

Author Response

Response to Reviewer 1

Dear Editor and Reviewers,

Thank you for your valuable comments and suggestions on our manuscript. We have sent the manuscript to Bullet Edits Company to improve the quality of the language. Enclosed is the certification of language editing. Please see the attachment.

We deeply appreciate your consideration of our manuscript. If you have any queries, please do not hesitate to contact us.

Thank you and best regards.

Sincerely yours,

Liling Tang

PhD, Professor

Reviewer 2 Report

Authors claim that they ordered the language correction service, but there's no confirmation. Please check if awful English was corrected before printing.

Now the review looks much better (readable, and comprehensive).

Author Response

Response to Reviewer 2

Dear Editor and Reviewers,

Thank you for your useful comments and suggestions on our manuscript. We have modified the manuscript accordingly as follow:

Comment 1: Authors claim that they ordered the language correction service, but there's no confirmation. Please check if awful English was corrected before printing.

Response: Thank you for your valuable comment. We apologize for the grammar and language issues presented in the review. We once again carefully checked the entire manuscript for grammatical errors according to your suggestions and have sent the manuscript to Bullet Edits Company to improve the quality of the language. Enclosed is the certification of language editing. Please see the attachment.

Finally, we thank you once again for your valuable comments and suggestions. We deeply appreciate your consideration of our manuscript. If you have any queries, please do not hesitate to contact us.

Thank you and best regards.

Sincerely yours,

Liling Tang

PhD, Professor
